# Long-Term Results of Proton Therapy for Hepatocellular Carcinoma Using Four-Dimensional Computed Tomography Planning without Fiducial Markers

**DOI:** 10.3390/cancers14235842

**Published:** 2022-11-26

**Authors:** Sayuri Bou, Shigeyuki Takamatsu, Sae Matsumoto, Satoko Asahi, Hitoshi Tatebe, Yoshitaka Sato, Mariko Kawamura, Satoshi Shibata, Tamaki Kondou, Yuji Tameshige, Yoshikazu Maeda, Makoto Sasaki, Kazutaka Yamamoto, Hajime Sunagozaka, Hiroyuki Aoyagi, Hiroyasu Tamamura, Satoshi Kobayashi, Toshifumi Gabata

**Affiliations:** 1Department of Radiology, Kanazawa University, Kanazawa 920-8641, Japan; 2Proton Therapy Center, Fukui Prefectural Hospital, Fukui 910-8526, Japan; 3Department of Radiology, Nagoya University Graduate School of Medicine, Nagoya 466-8550, Japan; 4Department of Diagnostic and Therapeutic Radiology, Kanazawa Medical University, Kahoku 920-0293, Japan; 5Department of Gastroenterology, Fukui Prefectural Hospital, Fukui 910-8526, Japan

**Keywords:** hepatocellular carcinoma, proton therapy, 4-dimensional CT planning, respiratory-gated irradiation, long term outcome

## Abstract

**Simple Summary:**

We investigated the long-term outcomes of patients with hepatocellular carcinoma (HCC) who underwent image-guided respiratory-gated proton therapy (IGPT) without fiducial markers with 4D-CT planning. IGPT achieved good local control, increased overall survival, and caused less severe toxicity. Our results suggest that less invasive IGPT for HCC is safe and effective without causing severe complications.

**Abstract:**

We report here the long-term results of marker-less respiratory-gated proton therapy (PT), without fiducial markers for hepatocellular carcinoma (HCC), which was planned using a four-dimensional computed tomography technique. Local tumor control (LTC) and overall survival (OS) were estimated using the Kaplan–Meier method. Toxicity was graded per CTCAE v5.0. Patients (n = 105; median age 73 years, range 38–90 years) with 128 lesions were treated. The median radiation dose was 66 gray relative biological effectiveness (GyRBE) (range, 52.8–82.5 GyRBE) delivered in 2.0 to 6.6 GyRBE fractions, depending on lesion volume, the involved liver, and the patient’s condition. The median follow-up of surviving patients was 63 months (range, 1–126 months), and the 5-year LTC and OS rates were 93.2% and 40.4%, respectively. Univariate and multivariate analyses identified tumors near the gastrointestinal tract as an independent risk factor for local recurrence and revealed that hepatic reserve, tumor stage, performance status, operability, sex, and portal vein thrombosis were independent risk factors for OS. Acute and late treatment-related grade 3 toxicities were experienced by eight patients (7.6%). Adverse events ≥ grade 4 were not evident. Marker-less respiratory-gated PT for HCC is a safe and effective treatment without severe complications.

## 1. Introduction

Liver cancer represents the third leading cause of cancer-related deaths worldwide [1]. Hepatocellular carcinoma (HCC) is the most common liver cancer, and the most important issue facing oncologists is its effective control. Liver transplantation, surgical resection, and radiofrequency ablation (RFA) are considered curative treatments for patients with sufficient liver function and a small tumor volume [2]. However, if a patient is not a candidate for surgery or RFA, an alternative treatment strategy is needed. Because of advances in radiation therapy technology, external beam radiotherapy (EBRT) using photon and particle beams is now regarded a rational treatment option for patients with localized HCC [3,4].

Proton therapy (PT) contributes to the improvement of EBRT for HCC [5,6,7]. When performing EBRT for HCC, it is important to confirm the location of the tumor and irradiate it at the correct location. Furthermore, breathing changes the position of the tumor caused by movement of the liver, and countermeasures are required [8]. In PT, the development of image-guided technology is delayed compared with photon therapy employing on-board cone-beam computed tomography (CT) imaging [9].

A respiratory PT-gating system with fiducial markers, developed for high-accuracy irradiation, achieves safe and effective treatment [7,10]. Therefore, a fiducial marker is placed in the liver to indicate the location of the tumor for classical PT with electronic portal imaging devices using a two-dimensional digital radiographic image to verify the position of the tumor relative to the patient’s anatomy [6,7]. However, patients with a cirrhotic liver may be at increased risk during the marker placement procedure because of pancytopenia [11].

To achieve less invasive PT, we used four-dimensional CT (4D-CT) planning for administering high-accuracy radiotherapy [12] and performed respiratory-gated PT without fiducial markers. In our two previous studies, we performed proton therapy using markerless registration for large liver cancers of 5 cm or larger [13] and those adjacent to the gastrointestinal tract [14], and we reported that the local control rate and degree of side effects were equivalent to treatment using conventional markers. Subsequently, we added case numbers and summarized the data on the long-term outcomes of markerless-registered proton therapy for liver cancer proximal to the gastrointestinal tract. Here, we report for the first time the long-term clinical outcomes achieved using this strategy.

## 2. Materials and Methods

The ethics committee of Fukui Prefectural Hospital approved this study (approval #14-09). Written informed consent was waived because of the study’s retrospective nature. The inclusion criteria were as follows: (1) a medically inoperable condition considered difficult to control with radiofrequency ablation (RFA) or the patient’s refusal to undergo surgery or RFA, (2) Eastern Cooperative Oncology Group performance status (PS) from 0 to 2, (3) hepatic function characterized by a Child–Pugh score ≤ 10, (4) no uncontrolled ascites, and (5) no extrahepatic metastasis. Consecutive patients considered eligible for analysis were selected between March 2011 and June 2017.

HCC was pathologically or clinically diagnosed using CT, magnetic resonance imaging (MRI), or both, as well as using serum levels of the tumor markers alpha-fetoprotein and des-gamma-carboxy prothrombin. Details of patient immobilization, simulation CT, target delineation, treatment planning techniques, and dose coverage goals were previously reported [13,14]. The treatment plan was made using a proton treatment planning system (XiO^®^-N, Elekta Corp., Stockholm, Sweden) in which the proton dose calculation was performed based on the pencil beam algorithm, and the proton dose distribution was formed to the target shape based on the passive scattering method using the patient’s collimator or multi-leaf collimators and patient’s bolus. CT images were obtained, and target contours were drawn at the end of the exhalation phase of the 4D-CT image. First, gross tumor volume (GTV) was manually delineated using contrast-enhanced CT and MRI. A clinical target volume (CTV) was drawn with a margin of 0.5 cm in all directions of the GTV. Respiratory-induced movements of target were analyzed by 4D-CT. ITV was determined as CTV plus an additional margin due to respiratory motion calculated by 4D-CT analysis. Internal respiratory motion margins were customized based on the amount of tumor motion visualized with a gate width of 1 s at end-tidal duty cycles of 17–25%. The internal target volume was determined as the clinical target volume plus a 5–10 mm margin. Proton treatment beams were controlled for respiratory-gated delivery at 17–25% duty cycle around end-expiration. A PBT beam was applied for approximately 1 s at the end of expiration. Respiratory gating was configured for a 1-s end-tidal centered beam-on cycle, and beam delivery was automated. The planning target volume was determined as the clinical target volume plus a 5 mm margin in all directions.

The setup at our facility involves a two-step procedure. It begins with biplane kVp X-rays taken in the anterior–posterior and lateral directions to align the vertebral bodies. The real-time movement of the diaphragm is observed using kV fluoroscopy. The diaphragm position at end of expiration phase is taken before treatment, and the distance from the planned diaphragm position is measured to determine the amount of deviation. The position of the diaphragm is corrected by moving the couch in the craniocaudal direction.

Various treatment protocols that had been used before February 2016 were combined, and since February 2016, treatment has been performed under a unified protocol throughout Japan. Typically, doses are delivered as follows: A total dose of 66 gray relative biological effectiveness (GyRBE) in 10 Fractions(Fr) was selected for tumors located in peripheral segments of the liver, 72.6–76 GyRBE in 20–22 Fr for tumors within 2 cm of the porta hepatis, and 74–76 GyRBE/2–2.2 GyRBE per fraction for tumors located adjacent to the gastrointestinal (GI) tract, using an irradiation schedule of 5 Fr per week.

Each fractional dose administered to all patients was configured using images acquired using a kV electronic portal imaging device (EPID) with reference to the vertebral bodies. Before daily treatment, EPID images of the diaphragm during the breath-holding exhale phase were acquired. The difference in the diaphragm position relative to the vertebral body was adjusted by comparing EPID images [15].

Patients were followed weekly during treatment and afterwards at approximately 3-month intervals. The National Cancer Institute Common Terminology Criteria for Adverse Events (CTCAE) version 5.0 was used for the assessment of toxicity. Acute toxicity was defined as occurring during PT or up to 6 months after completion of PT, and late toxicity was defined as occurring at any subsequent time. All endpoints were calculated from the day PT started.

The intervals for survival and local recurrence were calculated from initiating PT to the onset of the event. The Kaplan–Meier method was used to estimate overall survival (OS), progression free survival (PFS), and local tumor control (LTC), and a log-rank test was performed to evaluate the significance of differences between the two categories. A significant difference is indicated by *p* < 0.05. Significant variables identified using univariate analysis were selected and included in the stepwise selection procedure of multivariate analysis. Univariate and multivariate analyses using Cox proportional hazards regression models identified independent risk factors that predicted LTC and OS rates. Differences indicated by *p* < 0.05 were considered significant, and variables with *p* < 0.30 were entered into a multivariate analysis using a Cox proportional hazards model.

Statistical computations were performed using EZR (Saitama Medical Center, Jichi Medical University, Saitama, Japan), which is a graphical user interface for R (The R Foundation for Computing, Vienna, Austria). More precisely, EZR is a modified version of R commander designed to add statistical functions frequently used in biostatistics [16].

## 3. Results

### 3.1. Patients’ Characteristics

The baseline characteristics of 105 patients with 128 lesions are shown in Table 1 and Table 2. The median lesion size was 30 mm (range, 8–130 mm). The median radiation dose was 66 GyRBE (range, 52.8–82.5 GyRBE) delivered in 2.0 to 6.6 GyRBE fractions, depending on the lesion volume, the involved liver, and the patient’s condition. Nine protocols for respiratory-gated proton therapy (52.8–80.0 GyRBE in 10–38 Fr using 150-,190-, or 230-MeV proton beams) were implemented during the study period using an irradiation schedule of five fractions per week (Table 3).

Four patients failed to complete the planned protocol. Two of the four patients who developed uncontrollable ascites (grade 3) due to deterioration of the hepatic reserve or tumor growth completed PT earlier than stated in the planned protocol (52.8 GyRBE in 24 Fr or 57 GyRBE in 15 Fr). One of these four patients required blood transfusion treatment for gastrointestinal bleeding resulting from a gastric ulcer (grade 3), and PT was completed earlier than stated in the planned protocol (70 GyRBE in 35 Fr). One patient’s treatment was interrupted because of a cerebral hemorrhage (58 GyRBE in 29 Fr). These four patients were not excluded from analysis to accurately reflect the treatment results. For lesions close to the GI tract, some patients were replanned using the cone-down technique, and some underwent unexpected replanning because of an unpredictable change in the GI tract position or shape at the time [14].

We used a protocol of 74–76 GyRBE/2–2.2 GyRBE per fraction for tumors adjacent to the GI. Sixteen lesions (12%) were finally treated in this group. In this group, treatment was discontinued in three patients with large tumor sizes or with portal vein tumor thrombi. We have determined that a shorter treatment protocol rather than a longer treatment protocol may be preferable for such these cases. These cases may have required an early therapeutic effect. Even for tumors close to the GI tract, we used a single dose of 3.8 GyRBE, if early therapeutic effect is desired.

### 3.2. Treatment Results

The median follow-up of the 105 patients was 36 months (range, 1–126 months) and that of surviving patients was 63 months. The rates of LTC were 93.2% (95%CI: 86.2–96.7) at 3 years and 93.2% at 5 years (95%CI: 86.2–96.7). The rates of PFS were 35.4% (95%CI: 25.6–45.4) at 3 years and 26.2% at 5 years (95%CI: 17.3–36.0). Actuarial rates of OS were 56.9% (95%CI: 46.5–66.0) at 3 years and 40.4% (95%CI: 30.4–50.0) at 5 years (Figure 1 and Figure 2).

Table 4 shows the results of univariate and multivariate analyses of factors potentially associated with OS and LTC. Among all potential prognostic factors included in univariate analysis, sex (female), PS, hepatic reserve, operability, history of treatment, tumor characteristics (size, volume, thrombus, and staging), and irradiation protocol were significantly associated with OS. Multivariate analyses identified sex (female), Karnofsky Performance Status, operability, Tumor staging of the Union for International Cancer Control 8th edition, TNM staging of the Union for International Cancer Control 8th edition, and Barcelona Clinic Liver Cancer staging as significantly associated with prolonged OS. Univariate and multivariate analyses identified the lesion adjacent to the GI tract as the only factor significantly associated with LTC.

The rate of LTC far away from the GI tract was 96.9% (95%CI: 88.0–99.2) at 3 years and 5 years. The rate of LTC adjacent to the GI tract was 85.3% (95%CI: 67.8–93.7) at 3 years and 5 years.

### 3.3. Toxicity

Eight patients (7.6%) developed grade 3 side effects. As for acute toxicity, three patients (2.9%) developed grade 3 uncontrollable ascites. Among three patients, one experienced grade 3 hyperbilirubinemia. Ascites and hyperbilirubinemia occurred through portal vein thrombosis resulting from primary tumor progression. Among three patients with uncontrollable ascites, one completed PT and two failed to complete the planned protocol. One experienced a bleeding gastric ulcer (grade 3) that was treated using argon plasma coagulation. One experienced a cerebral hemorrhage (grade 3). As late toxicity, three patients (2.9%) developed grade 3 pleural effusion. Among patients who experienced grade 2 acute toxicity, six patients (5.7%) experienced grade 2 dermatitis. Among, patients who experienced grade-2 late toxicity, five experienced a rib fracture with pain, one suffered chest-wall pain without a rib fracture, one experienced radiation pneumonitis, and one suffered a colonic hemorrhage. Grade 4 or 5 toxicity was not noted during the acute or late observation periods.

## 4. Discussion

Through advances in radiation therapy technology, EBRT is considered a rational option for patients with localized HCC; and PT in particular contributes to improving EBRT for HCC [5,6,7]. As an example, Table 5 lists numerous cases of good LTC and OS using PT for HCC [4,5,6,7,17,18,19,20,21,22,23,24]. Here, we performed respiratory-gated PT without a reference marker for administering minimally invasive treatment. We used respiratory synchronized 4D-CT to plan treatment, which analyzed the respiratory movements of the target and proved useful for high-precision radiotherapy [10]. Targets were contoured at the end-expiratory phase using 4D-CT. Oshiro et al. reported that localization of the targets achieved greater reproducibility in end-expiration than that in end-inspiration [25]. PT is sensitive to displacement of the target, and liver tumors are easily moved because of respiration [8]. Therefore, fiducial markers are often placed in the liver to indicate the tumor location. Insertion of fiducial markers is reported to be safe and side effects are infrequent; however, patients with liver dysfunction may be at increased risk of adverse effects caused by the marker placement procedure.

When fiducial markers are not employed at our institute, we direct biplane kVp X-rays to the anterior-posterior and right-left directions using a robotic treatment table with six-axes corrections of movement, and the vertebral bodies are matched for correcting setup errors. Real-time movement of the diaphragm is monitored using kV X-ray fluoroscopy, such that the distances between the diaphragm positions at planning and during the peak-expiratory phase upon daily gating are measured to determine the amount of deviation. The position of the diaphragm is corrected by moving the treatment table toward the cranio-caudal (CC) direction. As stated above, the vertebral bodies and diaphragm are used as a landmark when radiation is delivered without a reference marker in the liver [8]. For example, Dawson et al. reported the diaphragm position correlates within approximately 0.2 cm with microcoils implanted near hepatic tumors and that the diaphragm likely correlates well with liver tumors [26]. Furthermore, Shimizu et al. found that the extents of respiratory movements of the liver varies depending on the segment and that surgery suppresses respiratory movements [27]. Therefore, it is important to confirm the movement using 4DCT and set the ITV margin in each case and individual lesion.

Here, we used a respiratory gating system controlled by monitoring abdominal wall motion with a laser sensor, in which breathing was 10–15 times/min, by setting the rhythm using a metronome. We used narrow gating (17–25%) of the duty cycle at end-expiration, which was approximately 1 s. Narrow gating prolonged the treatment times of patients with irregular respiration and minimized breathing movement. The gate width of our method was narrower than the commonly employed gate width [28]. It is therefore important for patients to undergo respiratory training to reduce irregular breathing patterns and treatment times [29]. At our facility, we provide pretreatment respiratory training of patient for 30–60 min.

Although this is a retrospective study, and an accurate comparison of treatment outcomes is not possible, in this study the 5-year OS rate was 40%, which is slightly lower than that observed in others. Iwata et al. prospectively investigated the outcomes of image-guided PT for operable or radiofrequency ablation-treatable primary HCC and found that the 5-year OS rate was 70% [24]. Fukuda et al. investigated the clinical outcomes of PT for HCC by Barcelona Clinic Liver Cancer (BCLC) staging [7]. The 5-year OS rates were 69% for patients with 0/A stage disease, 66% for patients with B stage disease, and 25% for patients with C stage disease. In this study, as many as 66% of cases were not indicated for surgery, and the higher proportion of group C stage BCLC disease in this study may have contributed to the poor OS results. These findings indicate that background factors such as liver function and the number and size of liver tumors may explain this difference in 5-year OS. In the current study, the 3- and 5-year LTC rates were 93.2% and 93.2%, respectively (Figure 1). Each protocol achieved good local control, and these results are consistent with the recent findings of studies using fiducial markers. There were two cases with portal vein thrombosis that led to liver transplantation after PT. It is thought that PT for HCC can suppress the progression of the disease, maintain a good general condition, and lead to transplantation. We believe that proton therapy will be a new multidisciplinary treatment for large-sized liver cancer.

Here, we found that five patients (4.8%) developed acute grade-3 adverse events, grade3 late adverse effects were experienced by three patients (3.8%), although ≥grade-4 toxicities were not observed. Komatsu et al. reported that grade-3 toxicities were experienced by seven (2.9%) patients who were asymptomatic or healed through conservative management [4]. Furthermore, one patient (0.4%) experienced refractory skin ulcers, a grade4 toxicity that required skin transplantation, and Iwata et al. found that one patient (2.2%) developed elevated hepatobiliary enzyme levels [24]. As mentioned above, the incidences of adverse effects at our institution are comparable to those reported by others (Table 5); and we achieved the same treatment results using fiducial markers as previously reported. Our method can therefore serve as a useful treatment for patients with liver dysfunction to reduce invasion to the extent possible.

Univariate and multivariate analyses revealed potential prognostic factors that were significantly associated with prolonged OS including sex (female), Karnofsky Performance Status, Child-Pugh classification, mUICC TNM stage, and BCLC stage. Among background factors between males and females, females were older, had a poor general condition, and had a poor hepatic reserve in many cases, which may have contributed to the poor overall survival rate (Appendix A). These factors are consistent with previously reported factors (Appendix A) such as patients’ characteristics (performance status, hepatic function reserve), tumor characteristics (clinical staging and tumor characteristics including size, volume, vascular thrombosis, prior treatment, and tumor markers), treatment factors (dose-fractionation regimen and primary tumor response), and treatment responses (proton-induced hepatic insufficiency and primary tumor response). Furthermore, univariate and multivariate analyses of LC revealed that a lesion close to the GI tract was the only factor significantly associated with LC. We used the diaphragm as an index for alignment, but in cases of liver S5, 6 tumors, it is possible that the alignment error was larger than that of liver S7, 8 tumors because they are far from the diaphragm. Tumor location on the diaphragm may lead to differential accuracy of tumor targeting and hence dose coverage to different parts of the tumor. There is also the possibility of a larger dose-misalignment of the tumor in the S5,6, that, combined with the lower prescription dose, leads to worse LTC. This positional inconsistency could also have increased Gl toxicity, but this is not an indication from our results (only one case had grade 3 GI toxicity). We think that it is possible to treat it safely using our radiation dose constraints in the GI tract [14]. However, there are concerns that it may be too weak to cure the tumor.

Although evidence indicates that tumor size is a major risk factor for poor OS (Appendix A), the present study did not identify this variable. High-precision radiation therapy is required to irradiate the target and protect organs at risks (OARs). When the GI tract becomes an OAR, the risk can be reduced using our method of adjusting the prescribed dose [14]. This method reduces the irradiation dose delivered to the tumor, which is adjusted to avoid severe damage to the GI tract. Moreover, our results show the importance of ensuring the administration of a sufficient irradiation dose to the target. Furthermore, recently reported surgical spacer placement techniques are useful for unresectable HCC [30]. This technique enables delivering a high dose of irradiation to HCC tissues that avoids GI tract damage. However, it may be difficult to treat certain cases with poor hepatic reserve or when the lesion is large, requiring early treatment. Further clinical trials are therefore required to confirm our potential breakthrough for effectively and safely treating HCC located close to the GI tract. It should be noted that a limitation of this study is its retrospective design, and a prospective study will be required to confirm the validity of our findings.

## 5. Conclusions

We examined the long-term results of less invasive PT for HCC using 4D-CT planning without fiducial markers. The results of our treatment method were comparable to those of the conventional method. This method is a safe and effective treatment without severe complications.

## Figures and Tables

**Figure 1 cancers-14-05842-f001:**
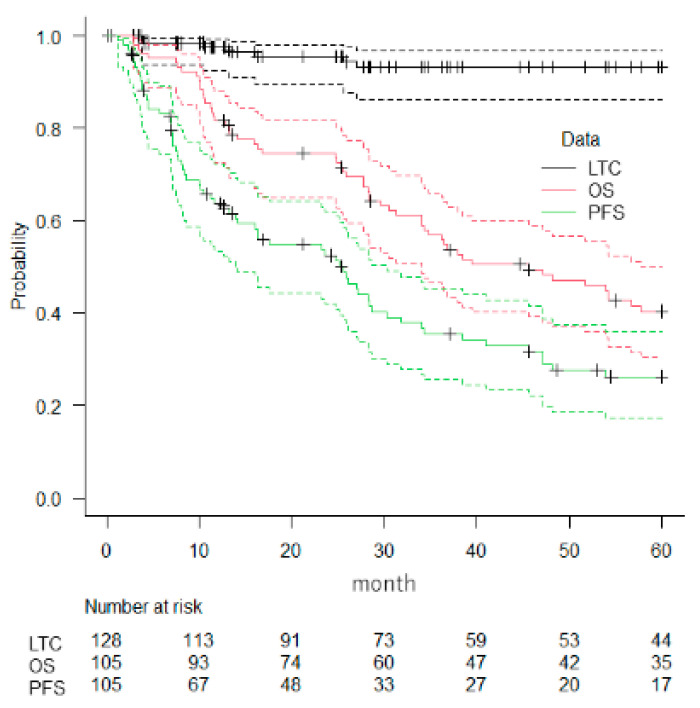
Survival analysis. Kaplan–Meier estimates of local tumor control (LTC), overall survival (OS), and progression free survival (PFS). Dotted lines represent 95% confidence intervals.

**Figure 2 cancers-14-05842-f002:**
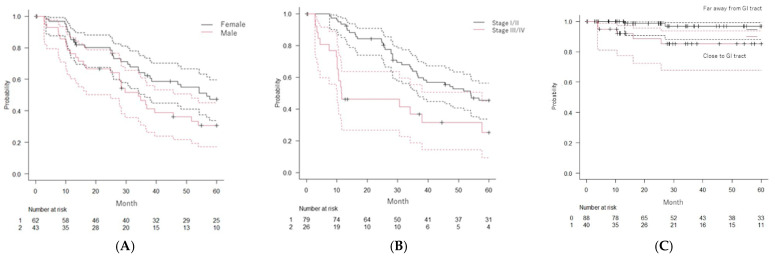
Kaplan–Meier estimates of overall survival (OS) according to (**A**) sex, and (**B**) UICC-stage. (**C**) Kaplan–Meier estimates LTC according to HCC located within 2 cm of the gastro-intestinal (GI) tract. Dotted lines represent 95% confidence intervals.

**Table 1 cancers-14-05842-t001:** Baseline characteristics of 105 patients who underwent image-guided proton therapy for hepatocellular carcinoma.

Patients/Lesions	105/128
Follow-up period, months	Median 36 (1–126) Surviving patients 63 (1–126)
Age, years	Median 73 (38–90)
Male/female	62 (59%)/43 (41%)
PS (0/1/2)	27 (26%)/69 (66%)/9 (8%)
KPS (100/90/80/70/60/50)	26 (25%)/68 (65%)/2 (2%)/6 (5%)/3 (3%)
Inoperable/refused surgery	69 (66%)/36 (34%)
Child Pugh A/B/C	87 (83%)/17 (16%)/1 (1%)
BCLC state (0/A/B/C)	2 (2%)/10 (10%)/13 (12%)/80 (76%)
ALBI score 1/2a/2b/3	48 (46%)/20 (19%)/33 (31%)/4 (4%)
Viral hepatitis +/−	70 (67%)/35 (33%)
HCV/HBV/both	57 (54%)/14 (13%)/1
Alcoholic LC/NASH/NBNC/IPH	13/6/8/1
UICC TNM: T1a/1b/2/3/4	17 (16%)/33 (31%)/31 (30%)/12 (11%)/12 (11%)
Prior treatment +/−	49 (47%)/56 (53%)
	TACE, 36; RFA, 16; PEIT, 5; Ope, 5; TAI, 1; PT, 1
AFP (ng/mL)	13 (1–625230)
PIVKAII (mAU/mL)	40 (11–72000)

Abbreviations: PS, Performance status; KPS, Karnofsky Performance Status; BCLC, Barcelona Clinic Liver Cancer staging; ALBI, albumin-bilirubin-grade; HCV, Hepatitis C virus; HBV, Hepatitis B virus; LC, Liver cirrhosis; NASH, Nonalcoholic Steatohepatitis; NBNC, etiology of non-B, non-C liver cirrhosis; IPH, Idiopathic portal hypertension; UICC TNM, TNM staging of the Union for International Cancer Control 8.0th edition; TACE, Trans-arterial Chemoembolization; RFA, radiofrequency ablation; PEIT, Percutaneous Ethanol Injection Treatment; Ope, operation; TAI, trans-arterial infusion; PT, proton therapy; AFP, Serum Alpha-fetoprotein levels; PIVKAII, Protein induced by vitamin K absence.

**Table 2 cancers-14-05842-t002:** Baseline tumor characteristics of 128 treated lesions.

Size, mm	Median 30 (9–140)
<4 cm/4–10 cm/>10 cm	84 (66%)/36 (28%)/8 (6%)
Pretreatment at PT site +/−	52 (41%)/76 (59%)
GI close < 2 cm/>2 cm	40 (31%)/88 (69%)
Organ type(number)	Eso(5); Stomach(10); Duodenum(7); Colon(21)
GTV (cm^3^)	14 (1–883)
PTV (cm^3^)	73 (11–1381)
Liver volume (cm^3^)	1152 (581–2196)

Abbreviations: PT, Proton therapy; GI close, close to gastrointestinal tract; Eso, esophagus, GTV, gross tumor volume; PTV, planning target volume.

**Table 3 cancers-14-05842-t003:** Treatment dose.

Treatment Dose	Dose Per Fraction	Lesion Number
66 GyRBE/10 Fr (BED10 109.5 Gy)	6.6 GyRBE	70 (55%)
76 GyRBE/20 Fr (BED10 104.9 Gy)	3.8 GyRBE	40 * (31%)
80.5 GyRBE/23 Fr (BED10 108.7 Gy)	3.5 GyRBE	1 (1%)
67.5 GyRBE/25 Fr (BED10 85.7 Gy)	2.7 GyRBE	1 (1%)
52.8–70.4 GyRBE/24–32 Fr (BED10 80.5–91.2 Gy)	2.2 GyRBE	8 * (6%)
58–76 GyRBE/29–38 Fr (BED10 80.5–91.2 Gy)	2.0 GyRBE	8 ** (6%)

* Includes one discontinued case; ** includes two discontinued cases. Abbreviations: GyRBE, gray relative biological equivalents; BED10, biologically effective dose with an alpha-beta ratio of 10.

**Table 4 cancers-14-05842-t004:** Univariate and multivariate analyses of clinical and treatment factors potentially associated with overall survival and local tumor control.

		Variables Affecting OS								Variables Affecting LTC							
		Univariate				Multiple				Univariate				Multiple			
Variables	(*n*)	*p*-Value	HR	95%CI		*p*-Value	HR	95%CI		*p*-Value	HR	95%CI		*p*-Value	HR	95%CI	
				Lower	Higher			Lower	Higher			Lower	Higher			Lower	Higher
Age	70>/70≤ (37/68)	0.727	0.918	0.5668	1.486	-	-	-	-	0.7015	1.340	0.2999	5.99	-	-	-	-
**Sex**	Male/Female (62/43)	**0.01894**	**0.570**	**0.3558**	**0.9114**	**0.0028**	**2.165**	**1.3040**	**3.5930**	0.6485	1.465	0.2840	7.5520	-	-	-	-
**PS**	0, 1/2, 3 (73/32)	**0.01145**	**0.483**	**0.2751**	**0.8492**	N.A	N.A	N.A	N.A	0.8468	0.851	0.1648	4.39	-	-	-	-
**KPS**	100, 90/80> (96/9)	**0.00003486**	**0.214**	**0.1033**	**0.4443**	**0.0066**	**0.344**	**0.1593**	**0.7431**	0.3481	0.362	0.0433	3.025	-	-	-	-
**CP class**	A/B,C (87/18)	**0.0005805**	**0.390**	**0.2282**	**0.6669**	N.A	N.A	N.A	N.A	0.9983	8,473,000	0.0000	Inf	-	-	-	-
**ALBI score**	1, 2a/2b, 3 (68/37)	**0.007123**	**0.527**	**0.3299**	**0.8400**	N.A	N.A	N.A	N.A	0.3396	2.807	0.3374	23.3600	-	-	-	-
Operable/inoperable	+/− (34/71)	**0.01291**	**0.525**	**0.3154**	**0.8723**	**0.012**	**0.491**	**0.2820**	**0.8551**	**0.2419**	**0.282**	**0.0339**	**Inf**	N.A	N.A	N.A	N.A
CLD	Yes/No (94/11)	0.3616	1.529	0.6142	3.805	-	-	-	-	0.9986	76,020,000	0.0000	Inf	-	-	-	-
Liver volume (mL)	<1150/1150≤ * (64/64)	0.543	0.868	0.5486	1.372	-	-	-	-	0.66	0.715	0.1599	3.194	-	-	-	-
**T stage**	**T1,2/T3,4** (80/25)	**0.01302**	**1.952**	**1.1510**	**3.31**	N.A	N.A	N.A	N.A	0.3292	2.267	0.4381	11.73	-	-	-	-
UICC	I, II/III, IV (79/26)	**0.001903**	**2.286**	**1.3570**	**3.853**	**0.0187**	**1.945**	**1.1170**	**3.384**	0.3204	2.308	0.4435	12.01	-	-	-	-
BCLC	0, A/B, C (93/12)	**0.003045**	**4.884**	**1.7110**	**13.94**	**0.0027**	**5.51**	**1.8070**	**16.83**	0.9211	0.899	0.1080	7.473	-	-	-	-
History of Pre-Tx	Y/N (48/57)	**0.06314**	**1.555**	**0.9761**	**2.476**	N.A	N.A	N.A	N.A	0.5564	1.568	0.3500	7.028	-	-	-	-
**TVT**	VP3,4/- (8/120)	**0.008378**	**2.708**	**1.2910**	**5.679**	N.A	N.A	N.A	N.A	**0.1867**	**4.175**	**0.5005**	**34.82**	N.A	N.A	N.A	N.A
**BED10**	<100/100 < (17/111)	**0.008435**	**0.425**	**0.2248**	**0.8033**	N.A	N.A	N.A	N.A	0.9985	79,150,000	1.1010	4.5800	-	-	-	-
**DPF**	<3/3 < (17/111)	**0.009518**	**2.323**	**1.2280**	**4.392**	N.A	N.A	N.A	N.A	0.7903	1.333	0.1603	11.08	-	-	-	-
**Tumor size**	**<7 cm/7 cm ≤** (113/15)	**0.03361**	**0.509**	**0.2728**	**0.9489**	N.A	N.A	N.A	N.A	0.9986	76,020,000	0.0000	Inf	-	-	-	-
GTV (mL)	<14/14 ≤ * (64/64)	**0.238**	**1.327**	**0.8293**	**2.125**	N.A	N.A	N.A	N.A	**0.205**	**2.889**	**0.5601**	**14.9**	N.A	N.A	N.A	N.A
PTV (mL)	<73/73 ≤ * (62/66)	**0.2773**	**1.304**	**0.8076**	**2.107**	N.A	N.A	N.A	N.A	0.6355	1.436	0.3213	6.421	-	-	-	-
Adjacent to the GI tract	<2 cm/2 cm ≤ (40/88)	0.4407	1.201	0.7512	1.929	-	-	-	-	**0.03702**	**5.729**	**1.1110**	**29.55**	**0.03702**	**5.729**	**1.1110**	**29.55**
Pre-Tx to PT site	Y/N (52/76)	0.1613	1.396	0.8752	2.227	-	-	-	-	0.7633	1.259	0.2810	5.642	-	-	-	-

* Cut-off value was the median value.Abbreviations: PS, Performance status; KPS, Karnofsky Performance Status; CP, Child-Pugh classification; ALBI, albumin-bilirubin-grade; BED10, biologically effective dose with an alpha-beta ratio of 10; DPF, dose per fraction; TVT, Tumor vascular thrombosis; CLD, chronic liver disease; T stage, Tumor classification of the Union for International Cancer Control stage 8th edition; UICC, TNM staging of the Union for International Cancer Control stage 8th edition; BCLC, Barcelona Clinic Liver Cancer staging; GI, gastrointestinal tract; GTV, gross tumor volume; PTV, planning target volume; Pre-Tx, Prior treatment; PT, proton therapy.

**Table 5 cancers-14-05842-t005:** Results of the present and other studies on proton therapy for HCC.

Author, Year	Study Type	Patient Number	Median Tumor Size (Range), mm	Treatment Dose	Fiducial Markers	Breathing Method	Timing of Irradiation	OS			LTC			Adverse Effects (Number of Patients)		Significant Risk Factors for OS		Significant Risk Factors for LC	
								3YR	4YR	5YR	3YR	4YR	5YR	Grade 3	Grade 4	Univariate	Multivariate	Univariate	Multivariate
**This study**	**Retro**	**105**	**30 (9–139)**	**52.8–80.5 Gy/10–38 Fr**	**None**	**Respiratory gating system ***	**Peak-expiratory phase**	**57%**	**48%**	**40%**	**93%**	**93%**	**93%**	**7.6% (8): ascites (3), pleural effusion (3), liver dysfunction(1), GI disorders(1),cerebral hemorrhage(1)**	**None**	**Sex, PS, KPS, CP, ALBI, T-stage, TVT, DPF, BED, mUICC, BCLC**	**Sex, KPS, CP, mUICC, BCLC**	**adjacent to the GI tract**	**adjacent to the GI tract**
Komatsu et al., 2011 [4]	Retro	242	NR	52.8–84.0 Gy/4–38 Fr	None	Respiratory gating system ***	Expiratory phase	NR	NR	38%	NR	NR	90%	2.9% (7): dermatitis(4), elevation of transaminase level(1), upper GI ulcer(1), biloma(1)	0.4%(1):Dermatitis	PS, CP, TVT, AFP, PIVKA II	PS, CP, TVT	Tumor size	Tumor size
Mizumoto et al., 2011 [6]	Retro	266	34 (6–130)	66.0–77.0 Gy/10–35 Fr	+	Respiratory gating system *	Expiratory phase	61%	NR	48%	87%	NR	81%	1.5% (4): dermatitis(1), perforation, bleeding or inflammation of the digestive tract(3)	None	**KPS**, CP, CTV, Normal liver volume, Tumor size, Prior treatment, Number of tumor	CP, CTV, Prior treatment (out-field)	None	None
Bush et al., 2011 [17]	Pros	76	55	63.0 Gy/15 Fr	None	Patients held their breaths voluntarily	Under breath-hold conditions.	2YR 25% 3YR 10%			NR			None	None	Tumor size, MELD, Milan criteria, San Francisco criteria	Milan criteria	NR	NR
Kawashima et al., 2011 [18]	Pros	60	45 (20–100)	60.0–76.0 Gy/10–26 Fr	+	Respiratory gating system ***	Expiratory phase	56%	NR	25%	90%	NR	86%	1.7% (1): hemorrhagic ulcer at the ascending colon	None	ICG, CP, PHI	CP, PHI, Prior treatment	NR	NR
Nakayama et al., 2011 [19]	Retro	47	NR	72.6–77 GyE/22–35 Fr	+	Respiratory gating system *	Expiratory phase	50%	34%	NR	88%	88%	NR	2.1% (1): colon ulcer with bleeding	None	NR	PS, CP, tumor size	NR	NR
Fukuda et al., 2017 [7]	Retro	129	39 (10–135)	66.0–77.0 Gy/10–35 Fr	+	Respiratory gating system ***	NR	NR	NR	25–69%	NR	NR	75–94%	None	None	NR	PS	NR	None
Kim et al., 2019 [5]	Retro	243	22 (10–170)	50.0–66.0 Gy/10 Fr	None	Respiratory gating system **	Expiratory phase	NR	NR	48%	NR	NR	88%	0.4% (1): Late GI toxicities, defined as gastric or duodenal ulcers	None	CP, AFP, Tumor size, TVT, mUICC, BCLC, Pre-Tx to PT site, Concurrent Tx, Post-Tx to PT site, Dose-fractionation regimens, Tumor response	CP, AFP level, mUICC, Dose-fractionation regimens, Primary tumor response	NR	NR
Chadlha et al., 2019 [20]	Retro	46	60 (15–210)	58–67.5 Gy/15–20 Fr	+/None	Respiratory gating or breath-hold technique or Free breathing	Free breathing or breathhold technique	1YR 73% 2YR 62%			NR			19.6% (9): diarrhea(1), erythema(1), ascites(nonmalignant)(4), hyperbilirubinemia(2), upper GI bleeding(1)	None	BED	BED, Prior treatment	Tumor size	None
Parzen et al., 2020 [21]	Pros	30	43 (12–94)	32.5–75.0 Gy/5–25 Fr	NR	NR	NR	1YR 72%			1YR 91%			3.3%(1): back pain	3.3%(1):G4 hyperbilirubinemia	NR	NR	None	NR
Kim et al., 2020 [22]	Pros	45	16 (10–68)	70.0 Gy/10 Fr	None	Respiratory gating system **	Expiratory phase	86%	86%	NR	95%	95%	NR	None	None	None	NR	None	NR
Bhangoo et al., 2021 [23]	Retro	37	50 (30–80)	37.5–67.5 Gy/5–15 Fr	+	Respiratory gating or breath-hold technique or Free breathing	NR	1YR 78%			1YR 94%			3%(1): acute toxicity (pain)	None	NR	NR	NR	NR
Iwata et al., 2021 [24]	Pros	45	25 (10–100)	66.0–72.6 Gy/10–22 Fr	+	Respiratory gating system *	End-expiration	NR	NR	70%	NR	NR	92%	2.2%(1): hepatobiliary enzyme elevation	None	Age, CP	BCLC	NR	NR

*; Respiratory gating system (Anzai Medical Co.) **; Respiratory gating system (Real-time position management (RPM) ***; Respiratory gating system (machine:NR).Abbreviations: OS, Overall survival; LTC, Local tumor control; YR, Year; Retro, Retrospective; Pros, Prospective; Gy, gray relative biological effectiveness; NR, not reported; GI, Gastro-intestinal; PS, Performance status; KPS, Karnofsky Performance Status; CP, Child-Pugh classification; ALBI, albumin-bilirubin-grade; TVT, Tumor vascular thrombosis; DPF, dose per fraction; BED, Biological equivalent dose; mUICC, TNMstage; BCLC, Barcelona Clinic Liver Cancer staging; AFP, Serum Alpha-fetoprotein levels; PIVKA II, serum PIVKA II level; CTV, clinical target volume; MELD, model of end-stage liver disease; ICG, Pretreatment ICG R15; PHI, Proton-induced hepatic in sufficiency; Pre-Tx, Prior treatment; PT, proton therapy; Concurrent Tx, concurrent treatment; Post-Tx, post treatment.

## Data Availability

The data presented in this study are available on request from the corresponding author. The data are not publicly available due to institutional guidelines.

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
