# Peer review of "Long-Term Results of Proton Therapy for Hepatocellular Carcinoma Using Four-Dimensional Computed Tomography Planning without Fiducial Markers"

_cancers, 2022, doi:10.3390/cancers14235842_

Round 1
Reviewer 1 Report
This retrospective study investigated the clinical outcomes after proton beam therapy (PBT) with respiratory gating technique in hepatocellular carcinoma and analyzed associated prognostic factors. While the results of this study are potentially interesting and informative, some critical points need to be clarified. I have the following suggestions and criticisms.
#1. Because a number of previous PBT studies for HCC is already demonstrated the efficacy of PBT, I’d like to ask what is the originality of this study.
#2. In Table 5, the incidence of severe adverse event is relatively high in the present study (> 5%) compared to other studies. I’m afraid that set-up reproducibility of less invasive technique can be worse and it can lead more toxicity. What is the authors opinion?
#3. Inclusion criteria and information of other treatment need to be clarified. Is recurrent ds or treatment naïve ds? or the association with previous treatment and PBT is not clear, that is it combined therapy or salvage PBT to local recurrence after previous treatment. And the information regarding post-PBT treatment is limited such as administration of systemic treatment.
#4. In Table 4, please provide the number of patients for each categorization.
#5. Please provide the description which respiratory phase does PBT beam on during treatment in Methods.
Author Response
Reviewer #1
We wish to express our appreciation to the Reviewer for his or her insightful comments, which have helped us significantly improve the paper. We are thankful for the time and energy you expended.
Our responses to the referees’ comments are as follows:
#1. Because a number of previous PBT studies for HCC is already demonstrated the efficacy of PBT, I’d like to ask what is the originality of this study.
Answer: You have raised an important point.
In our two previous studies, we performed proton therapy using markerless registration for large liver cancers of 5 cm or larger (Shibata, S. et al. Cancers 2018; 10: 71.) and those adjacent to the gastrointestinal tract (Mizuhata, M. et al. Cancers 2018; 10: 58.), and we reported that the local control rate and degree of side effects were equivalent to treatment using conventional markers. Subsequently, we added case numbers and summarized the data on the long-term outcomes of markerless-registered proton therapy for liver cancer proximal to the gastrointestinal tract.
Marker placement can be safely performed, but we investigated the feasibility of markerless treatment to reduce invasiveness. We obtained results that demonstrated that both the locality control rate and the side effects were equivalent to those of registration using conventional markers. Markerless treatment eliminates the need for invasive procedures associated with treatment, reducing the burden on patients.
In addition, the ASTRO guidelines 2022 (Apisarnthanarax et al. Practical Radiation Oncology 2022) list the refinement of optimal tumor dose and normal liver dose constraints as an important future research topic to clarify the role of proton therapy (radiation therapy) in liver cancer treatment. In addition to considering the prescribed dose based on the degree of liver dysfunction, it is also necessary to pay attention to the risk according to the location and size of the tumor. Furthermore, it seems necessary to stratify cases and administer appropriate prescription doses. In this study, we summarized the long-term course of proton therapy for tumors close to the gastrointestinal tract and obtained results of high local control rate and low incidence of side effects with the prescribed dose and treatment method. We believe that this data will serve as a reference for establishing a path for future liver cancer treatment. These are the key points and novelty of this paper.
#2. In Table 5, the incidence of severe adverse event is relatively high in the present study (> 5%) compared to other studies. I’m afraid that set-up reproducibility of less invasive technique can be worse and it can lead more toxicity. What is the authors opinion?
Answer: This is an important point.
In the same markerless reports (Kim et al. 2019, Komatsu et al. 2011) as ours, there were no reports of grade 3 or higher early adverse events, but in the study by Kim et al., one person (2.5%) had gastric ulcer or duodenal ulcer as a grade 3 or higher late adverse event. Komatsu et al. (2011) reported that grade 3 or higher late adverse events occurred in 8 of 242 patients (3%). Late adverse events occurred in three patients (2.9%) in our study, roughly in line with previous reports of approximately 3%. In addition, in the report of Chadoha et al. 2019, which examined mixed cases with markers and without markers, grade 3 or higher early adverse events occurred in six cases (13%). Of these, non-malignant ascites was found in four patients (9%). This was similar to our finding in that they recognized the early onset of ascites. In our study, three patients (2.9%) developed grade 3 ascites, which may not be particularly high compared with the above. Ascites is not only a side effect of proton therapy, but also a symptom that can be observed depending on the degree of liver dysfunction. Because many of the cases included in our study were judged to be difficult to operate on because of liver dysfunction, it is thought that some cases with near-limit liver function were included. The clinical features of these cases suggest that ascites may have developed from minimal progression of liver dysfunction.
Regarding cases in which markers were implanted, according to a report by Kawashima et al. 2011, early adverse events during irradiation were within a clinically insignificant range, but many, such as blood cell abnormalities, were observed in 23.3% of cases. A late adverse event requiring surgery was also observed in 1.7%. Nakayama et al. 2011 reported that late adverse events requiring surgery occurred in 2.1%.
In our study, four (3.8%) had early adverse events of grade 3 or higher and three (2.9%) had late adverse events. When evaluating early and late adverse events separately, it seemed that the incidence of side effects in our study was not outstanding. In addition, we did not identify any cases in which side effects of proton therapy led to the addition of surgical therapy or direct causes of death. At this stage, we cannot say that markerless therapy caused many of the side effects.
#3. Inclusion criteria and information of other treatment need to be clarified. Is recurrent ds or treatment naïve ds? or the association with previous treatment and PBT is not clear, that is it combined therapy or salvage PBT to local recurrence after previous treatment. And the information regarding post-PBT treatment is limited such as administration of systemic treatment.
Answer: Thank you for your insightful comment.
In this study, 53% of patients (56 cases) were treated with proton therapy as the initial treatment for HCC (treatment-naïve). Furthermore, 49 cases with 52 lesions (41%) were previously treated. This study was an evaluation of treatment outcomes at our facility, and only one patient had received proton therapy at another facility.
No patients received chemotherapy combined with proton beam therapy, and treatment was performed with only proton beam therapy. Presence or absence of treatment prior to proton therapy was not included in the eligibility criteria.
Basically, treatment aimed at radical cure rather than palliative irradiation was performed, but prognosis was shortened as a result in cases with advanced stage or poor liver reserve function.
#4. In Table 4, please provide the number of patients for each categorization.
Answer: We added the number of patients for each categorization to Table 4.
#5. Please provide the description which respiratory phase does PBT beam on during treatment in Methods.
Answer: Thank you for highlighting this. Proton treatment beams were controlled for respiratory-gated delivery at 17%–25% duty cycle around end-expiration. A PBT beam was applied for approximately 1 second at the end of expiration. Respiratory gating was configured for a 1-second end-tidal centered beam-on cycle, and beam delivery was automated.
Reviewer 2 Report
I was asked to review a retrospective clinical study concerning the outcomes of proton therapy of HCC. Generally, the study is sound and the manuscript is well written. Generally, I would like to recommend this paper for publication in Cancers. However, please note the following concerns:
· I wonder about the prominent mentioning of “less invasive” and “without fiducial markers”, e.g. in the title. I think both terms correspond to the same aspect and do not need duplication in the title. Furthermore, this aspect is neither novel nor unique (as Tab. 5 indicates). From my point of view, clinical users must implement a motion management technique The authors of the current study decided for quite common tools. So I don’t see a need to stress these aspects (e.g.) in the discussion. Generally, this does not diminish the quality of the study.
· It should be mentioned which delivery technique is used (double scattering, spot scanning?)
· The x-ray based positioning in the treatment room used the diaphragm and the vertebrae as a reference. I gather from the decription that the relative position of diaphragm and vertebrae could differ from the anatomy during the planning CT. Proton fields are quite sensitive to such anatomical changes. Did the authors conduct a robustness evaluation?
· P. 2, line 95: here it is described that a generic ITV margin is used. But the ITV should be constructed from the phases of the 4D-CT, shouldn’t it?
T Two previouos studies are cited (Refs. 13 & 14). On a first glance the content, methods and sometimes even the wording is similar to the current manuscript. I think the authors should make clear what the substantial improvement of the current study over previous ones is.
· P. 3, line 100: “FR” was not introduced.
· Tab. 4: the number of significant digits should be written coherently.
Author Response
We wish to express our appreciation to the Reviewer for his or her insightful comments, which have helped us significantly improve the paper. We are thankful for the time and energy you expended.
Our responses to the referees’ comments are as follows:
- I wonder about the prominent mentioning of “less invasive” and “without fiducial markers”, e.g. in the title. I think both terms correspond to the same aspect and do not need duplication in the title. Furthermore, this aspect is neither novel nor unique (as Tab. 5 indicates). From my point of view, clinical users must implement a motion management technique The authors of the current study decided for quite common tools. So I don’t see a need to stress these aspects (e.g.) in the discussion. Generally, this does not diminish the quality of the study.
Answer: We agree with your assessment.
We have changed the title to “Long-term results of proton therapy for hepatocellular carcinoma using four-dimensional computed tomography planning without fiducial markers”.
- It should be mentioned which delivery technique is used (double scattering, spot scanning?)
Answer: Thank you for your suggestion.
We have changed the sentence as follows:
The treatment plan was made using a proton treatment planning system (XiO®-N, Elekta Corp.Stockholm, Sweden) in which the proton dose calculation was performed based on the pencil beam algorithm, and the proton dose distribution was formed to the target shape based on the passive scattering method using the patient's collimator or multi-leaf collimators and patient's bolus.
- The x-ray based positioning in the treatment room used the diaphragm and the vertebrae as a reference. I gather from the decription that the relative position of diaphragm and vertebrae could differ from the anatomy during the planning CT. Proton fields are quite sensitive to such anatomical changes. Did the authors conduct a robustness evaluation?
Answer:
The setup at our facility involves a two-step procedure. It begins with biplane kVp X-rays taken in the anterior–posterior and lateral directions to align the vertebral bodies. The real-time movement of the diaphragm is observed using kV fluoroscopy. The diaphragm position at the maximum expiratory phase is taken before treatment, and the distance from the planned diaphragm position is measured to determine the amount of deviation. The position of the diaphragm is corrected by moving the couch in the craniocaudal direction. This is how our facility responds to inter-fractional baseline shifts. Intra-fractional diaphragmatic baseline shifts may also lead to errors in the proton beam path, but this can be addressed by setting the ITV margin, PTV margin, and MLC margin for each case and lesion at the time of treatment planning. We believe that minor shifts will be contained within the margins and will not amount to range errors. When a baseline falls outside the gating window, we usually wait to improve the baseline naturally. No cranio-caudal corrections were made to the couch during treatment in this study.
- P. 2, line 95: here it is described that a generic ITV margin is used. But the ITV should be constructed from the phases of the 4D-CT, shouldn’t it?
Answer: You have raised an important point. We calculated the tumor motion distance between the inspiratory phase and expiratory phase. Finally, the internal margin was customized based on the amount of respiratory-induced motion visualized in the gate width for 1 s; a duty cycle of approximately 20% was centered at end-expiration. The internal target volume was determined as the clinical target volume plus a 5–10 mm margin. A PBT beam was applied for 1 second at the end of expiration. Respiratory gating was configured for a 1-second end-tidal centered beam-on cycle, and beam delivery was automated.
T Two previouos studies are cited (Refs. 13 & 14). On a first glance the content, methods and sometimes even the wording is similar to the current manuscript. I think the authors should make clear what the substantial improvement of the current study over previous ones is.
Answer: You have raised an important point.
In our two previous studies, we performed proton therapy using markerless registration for large liver cancers of 5 cm or larger (Shibata, S. et al. Cancers 2018; 10: 71.) and those adjacent to the gastrointestinal tract (Mizuhata, M. et al. Cancers 2018; 10: 58.), and we reported that the local control rate and degree of side effects were equivalent to treatment using conventional markers. Subsequently, we added case numbers and summarized the data on the long-term outcomes of markerless-registered proton therapy for liver cancer proximal to the gastrointestinal tract.
All 40 cases evaluated in the paper of Mizuhata et al. and 29 patients in the paper of Shibata et al. are included in the current case.
We have added the following sentence to the Introduction:
In our two previous studies, we performed proton therapy using marker-less registration for large liver cancers 5 cm or larger (13) and those adjacent to the gastrointestinal tract (14), and we reported that the local control rate and degree of side effects were equivalent to treatment using conventional markers. Subsequently, we added case numbers and summarized data on the long-term outcomes of markerless-registered proton therapy for liver cancer proximal to the gastrointestinal tract.
- P. 3, line 100: “FR” was not introduced.
Answer: We have defined this abbreviation in the revised manuscript (Fr, fractions).
- Tab. 4: the number of significant digits should be written coherently
Answer: We have added this number to Table 4.
Reviewer 3 Report
This manuscript reported the clinical outcomes of hepatocellular carcinomas by proton therapy delivered by fiducial-free respiration-gating technique. Although the reproted results are promising, there are important questions that the authors have to clarify before the article can be considered for acceptance.
Major comments:
1. A portion of the patients seems to have been included in other studies (Shibata et al. Cancers 2018 and Mizuhata et al. Cancers 2018) by the same group of authors. However, this was not disclosed anywhere in the manuscript. Results reported in this manuscript were direct copies from the last studies; for example, p.4 line 159-161 "One of the four who did not complete the protocol had a gastric hemorrhage with an ulcer that was treated through transfusion (grade 3), and PT was completed earlier than stated in the planned protocol (70 GyRBE in35 Fr)" was almost the same results reported earlier by Mizuhata et al (Cancers 2018) "One of the two who did not complete the protocol had gastric hemorrhage with an ulcer treated by transfusion (grade 3), and PBT was finished earlier than the planned protocol (70 cobalt gray equivalents (CGE) in 35 Fr)". Toxicity outcomes (line 157-159 in p4) of "Two of the four patients who developed uncontrollable ascites (grade 3) completed PT earlier than stated in the planned protocol (52.8 GyRBE in 24 Fr or 57GyRBE in 15 Fr)" also seem to overlap with the earlier study by Mizuhata et al (Cancers 2018) " Two of the four patients who 157 developed uncontrollable ascites (grade 3) completed PT earlier than stated in the planned 158 protocol (52.8 GyRBE in 24 Fr or 57GyRBE in 15 Fr)". In the current study, there were 40 lesions < 2cm from the gastrointestines (GI). This was the exact number of patients in Mizuhata et al. (Cancers 2018) reporting the clinical outcomes of PT in HCC adjacent to GI. The reported local control rates were almost the same. Are these patients complete overlap in the two studies? This is extremely important for the authors to explicitly state how many patients from the earlier studies have been included in the current study. Otherwise, reporting on a large overlapped group of patients in multiple studies will attribute to artificially consistent results in the literatures.
2. how local control was defined? As the absence of PD within PTV as per mRECIST / RECISTv1.1?
3. The proximity to GI was found to be predictive factor of local tumor control (LTC). In the introduction, the authors stated that the unified dose prescription protocol of 74-76GyRBE/2-2.2GyRBE per fractions for tumors adjacent to the GI since Feb 2016. However, the authors did not describe the dose prescription approach to tumors adjacent to GI before Feb 2016. Were lower total and fraction doses, i.e., BED10 of 80.5-91.2Gy in Table 3 used in these patients? Was there a correlation between prescription dose and tumor proximity to GI and how was it handled in the multivariate analysis? The reviewer was also curious about the impact of the diaphragm-based tumor targeting on the dose coverage to tumor in the liver segment 5 and 6 that frequently abut the GI. Considering the differential motion of between liver segments V+VI and VII+VIII, basing tumor location on diaphragm may lead to differential accuracy of tumor targeting and hence dose coverage to different part of the tumor. Would there be a possibility of larger dose-misalignment of the tumor in the V+VI segment that combined with the lower prescription dose led to the worse LTC?
4. The patient cohort consists of Child-Pugh (CP) class A-C, with CP score (CPS) up to 10; Radiation therapy has been typically not considered for CP class B7+ due to the significant risk of classical/non-classical RILD. It would be much to include the CPS in class B to illustrate the safety of proton therapy in this vulnerable cohort. In addition, the study did not describe the dose fractionation approach and dose constraints tailored to patients with different CP class per the ASTRO guideline on external beam radiotherapy for primary liver cancers (Apisarnthanarax et al. Practical Radiation Oncology 2022)
5. The authors referred their treatment setup to previous study. It was unclear whether the setup was bone (vertebrae)-based or diaphragm-based in the fluoroscopic mode, or in a two-step procedure initially based on bony anatomy followed by alignment to the diaphragm. If the setup was a two-step approach, how the authors dealt with the interfractional diaphragm baseline drift. Since the treatment port was aligned according to the diaphragm position, the proton beam path could change quite substantially when there was an interfractional baseline drift. Similarly, when there was an intrafractional baseline shift in the diaphragm position that fell outside the “amplitude (or time) gating” window, as shown in the fluoroscopic monitoring, was it corrected for during the treatment by moving the treatment table crani-caudally? Additional information about the treatment setup / tumor targeting is essential to provide confidence of such fiducial-less respiration-gating treatment technique in connection with the clinical outcomes.
6. Proton therapy can be a viable option to large HCC because of possible dose escalation without excessive liver toxicity as demonstrated in this study and others, providing the opportunity of conversion therapy and bridge therapy after significant downstaging. It would be interesting to know if there were, and how many HCCs who were initially considered medically inoperable, were eventually converted to resectable. This information may provide important rationale to substantiate the administration of proton therapy to HCC.
Minor comments:
1. P2, line 59-60. “Furthermore, breathing changes the position of the tumor 59 caused by movement of the liver, …” grammar problem
2. P2, line 64, can delete “Therefore”
3. It said in p2. line 88-89 “The radiation treatments were designed using a radiation treat-88 ment planning system with a proton pencil beam algorithm” it seems to suggest that the proton therapy was delivered by pencil beam scanning technique. It would be important to specifically describe the type of proton therapy delivery technique was used.
4. In Table 2, it is recommended to the move Eso/stomach/duodenum/colon on the left column to the right column, e.g., Eso (5); stomach (10); duodenum (7), and so on.
5. Abbreviation “LC” liver cirrhosis in Table 1 was also used for local control. Consider abbreviating local control to local tumor control (LTC) to avoid confusion to the reader.
Author Response
We wish to express our appreciation to the Reviewer for his or her insightful comments, which have helped us significantly improve the paper. We are thankful for the time and energy you expended.
Our responses to the referees’ comments are as follows:
Major comments:
- A portion of the patients seems to have been included in other studies (Shibata et al. Cancers 2018 and Mizuhata et al. Cancers 2018) by the same group of authors. However, this was not disclosed anywhere in the manuscript. Results reported in this manuscript were direct copies from the last studies; for example, p.4 line 159-161 “One of the four who did not complete the protocol had a gastric hemorrhage with an ulcer that was treated through transfusion (grade 3), and PT was completed earlier than stated in the planned protocol (70 GyRBE in35 Fr)” was almost the same results reported earlier by Mizuhata et al (Cancers 2018) “One of the two who did not complete the protocol had gastric hemorrhage with an ulcer treated by transfusion (grade 3), and PBT was finished earlier than the planned protocol (70 cobalt gray equivalents (CGE) in 35 Fr)”. Toxicity outcomes (line 157-159 in p4) of “Two of the four patients who developed uncontrollable ascites (grade 3) completed PT earlier than stated in the planned protocol (52.8 GyRBE in 24 Fr or 57GyRBE in 15 Fr)” also seem to overlap with the earlier study by Mizuhata et al (Cancers 2018) “ Two of the four patients who 157 developed uncontrollable ascites (grade 3) completed PT earlier than stated in the planned 158 protocol (52.8 GyRBE in 24 Fr or 57GyRBE in 15 Fr)”. In the current study, there were 40 lesions < 2cm from the gastrointestines (GI). This was the exact number of patients in Mizuhata et al. (Cancers 2018) reporting the clinical outcomes of PT in HCC adjacent to GI. The reported local control rates were almost the same. Are these patients complete overlap in the two studies? This is extremely important for the authors to explicitly state how many patients from the earlier studies have been included in the current study. Otherwise, reporting on a large overlapped group of patients in multiple studies will attribute to artificially consistent results in the literatures.
Answer: In our two previous studies, we performed proton therapy using markerless registration for large liver cancers of 5 cm or larger (Shibata, S. et al. Cancers 2018; 10: 71.) and those adjacent to the gastrointestinal tract (Mizuhata, M. et al. Cancers 2018; 10: 58.), and we reported that the local control rate and degree of side effects were equivalent to treatment using conventional markers. Subsequently, we added case numbers and summarized the data on the long-term outcomes of markerless-registered proton therapy for liver cancer proximal to the gastrointestinal tract.
All 40 cases evaluated in the paper of Mizuhata et al. and 29 patients in the paper of Shibata et al. are included in the current study.
We have added the following sentences to the Introduction:
In our two previous studies, we performed proton therapy using marker-less registration for large liver cancers 5 cm or larger (13) and those adjacent to the gastrointestinal tract (14), and we reported that the local control rate and degree of side effects were equivalent to treatment using conventional markers. Subsequently, we added case numbers and summarized data on the long-term outcomes of markerless-registered proton therapy for liver cancer proximal to the gastrointestinal tract.
- how local control was defined? As the absence of PD within PTV as per mRECIST / RECISTv1.1?
Answer: Thank you for your comments. HCC after proton therapy often has reduced nodules, scars, and arterial enhancement of the tumor (Takamatsu S et al. Jpn J Radiol. 2018;36(4):241-256.). Therefore, when we use mRECIST/RECISTv1.1 criteria, most cases are finally judged as PR. Thus, to avoid this problem, we defined local recurrence as when the lesion re-enlarges following shrinkage after treatment.
- The proximity to GI was found to be predictive factor of local tumor control (LTC). In the introduction, the authors stated that the unified dose prescription protocol of 74-76GyRBE/2-2.2GyRBE per fractions for tumors adjacent to the GI since Feb 2016. However, the authors did not describe the dose prescription approach to tumors adjacent to GI before Feb 2016. Were lower total and fraction doses, i.e., BED10 of 80.5-91.2Gy in Table 3 used in these patients? Was there a correlation between prescription dose and tumor proximity to GI and how was it handled in the multivariate analysis?
The reviewer was also curious about the impact of the diaphragm-based tumor targeting on the dose coverage to tumor in the liver segment 5 and 6 that frequently abut the GI. Considering the differential motion of between liver segments V+VI and VII+VIII, basing tumor location on diaphragm may lead to differential accuracy of tumor targeting and hence dose coverage to different part of the tumor. Would there be a possibility of larger dose-misalignment of the tumor in the V+VI segment that combined with the lower prescription dose led to the worse LTC?
Answer: Before February 2016, we used the same protocol of 74–76 GyRBE/2–2.2 GyRBE per fraction for tumors adjacent to the GI. Finally, 16 lesions (12%) were treated in this protocol. As you have stated, these cases were the low BED group (BED10 of 80.5–91.2 Gy). In this group, treatment in three patients with a large tumor size or with portal vein tumor thrombi was discontinued.
Therefore, in such cases, if a rapid appearance of therapeutic effects was desirable, treatment was performed at 76 Gy/20 fractions (24 lesions,19% with GI near lesions).
We have added the following sentence to the revised manuscript:
We used a protocol of 74–76 GyRBE/2–2.2 GyRBE per fraction for tumors adjacent to the GI. Sixteen lesions (12%) were finally treated in this group. In this group, treatment was discontinued in three patients with large tumor sizes or with portal vein tumor thrombi. Therefore, in such cases, if a rapid appearance of therapeutic effects was desirable, treatment was performed at 76 Gy/20 fractions (24 lesions,19% with GI near lesions).
Our group previously reported that segments 5 and 6 of the liver can move a great deal, which is affected by abdominal surgical history, as you have suggested (Shimizu Y et al. Jpn J Radiol. 2018;36:511–518). Therefore, it is useful to check the motion of the tumor because we can change the margin for each patient using 4DCT planning. However, as you describe, there would be the possibility of a larger dose-misalignment of the tumor in the V+VI segment, that, combined with the lower prescription dose, leads to worse LTC. This positional inconsistency could also have increased gastrointestinal toxicity, but this is not an indication from our results and may be due to the safety of radiation dose constraints to the gastrointestinal tract (Mizuhata, M. et al. Cancers 2018; 10: 58.). However, there are concerns that it may be too weak in terms of tumor control.
We have added the following sentence to the Discussion:
We used the diaphragm as an index for alignment, but in cases of liver S5, 6 tumors, it is possible that the alignment error was larger than that of liver S7, 8 tumors because they are far from the diaphragm. Tumor location on the diaphragm may lead to differential accuracy of tumor targeting and hence dose coverage to different parts of the tumor. There is also the possibility of a larger dose-misalignment of the tumor in the V+VI segment, that, combined with the lower prescription dose, leads to worse LTC. This positional inconsistency could also have increased Gl toxicity, but this is not an indication from our results (only one case had grade 3 GI toxicity). We think that it is possible to treat it safely using our radiation dose constraints in the GI tract (Mizuhata, M. et al. Cancers 2018; 10: 58.). However, there are concerns that it may be too weak to cure the tumor.
- The patient cohort consists of Child-Pugh (CP) class A-C, with CP score (CPS) up to 10; Radiation therapy has been typically not considered for CP class B7+ due to the significant risk of classical/non-classical RILD. It would be much to include the CPS in class B to illustrate the safety of proton therapy in this vulnerable cohort. In addition, the study did not describe the dose fractionation approach and dose constraints tailored to patients with different CP class per the ASTRO guideline on external beam radiotherapy for primary liver cancers (Apisarnthanarax et al. Practical Radiation Oncology 2022)
Answer: The present study does not refer to the ASTRO guidelines 2022, and thus a dose fractionation approach and dose limitation by CP class were not used. We also did not attempt to further classify and evaluate the CP classes by score.
The ASTRO guidelines 2022 state that the total dose and dose per fraction should be selected differently for CP class A and B7 patients, especially considering the different normal liver dose constraints between these patients. This is because there was a difference in the exacerbation of liver dysfunction after irradiation. Furthermore, much of the data on hepatic decompensation after EBRT for HCC came from the SBRT literature, which largely excluded patients with CP class B (score 8) or C hepatic dysfunction, even in the same CP class B. Thus, the data of cases with a score of 8 or more are not reflected well, and data from cases with a score of 8 or higher in the CP class are solicited.
We think that it is very interesting that it may be possible to reduce the exacerbation of liver dysfunction by subdividing patients, not only by CP class but also by score, and administering an appropriate prescribed dose. In the future, we will subdivide by CP score and investigate changes in CP class and score after irradiation.
- The authors referred their treatment setup to previous study. It was unclear whether the setup was bone (vertebrae)-based or diaphragm-based in the fluoroscopic mode, or in a two-step procedure initially based on bony anatomy followed by alignment to the diaphragm. If the setup was a two-step approach, how the authors dealt with the interfractional diaphragm baseline drift. Since the treatment port was aligned according to the diaphragm position, the proton beam path could change quite substantially when there was an interfractional baseline drift. Similarly, when there was an intrafractional baseline shift in the diaphragm position that fell outside the “amplitude (or time) gating” window, as shown in the fluoroscopic monitoring, was it corrected for during the treatment by moving the treatment table crani-caudally? Additional information about the treatment setup / tumor targeting is essential to provide confidence of such fiducial-less respiration-gating treatment technique in connection with the clinical outcomes.
Answer: The setup at our facility involves a two-step procedure. It begins with biplane kVp X-rays taken in the anterior–posterior and lateral directions to align the vertebral bodies. The real-time movement of the diaphragm is observed using kV fluoroscopy. The diaphragm position at the maximum expiratory phase is taken before treatment and the distance from the planned diaphragm position is measured to determine the amount of deviation. The position of the diaphragm is corrected by moving the couch in the craniocaudal direction. This is how our facility responds to inter-fractional baseline shifts. Intra-fractional diaphragmatic baseline shifts may also lead to errors in the proton beam path, but this can be addressed by setting the ITV margin, PTV margin, and MLC margin for each case and lesion at the time of treatment planning. We believe that minor shifts will be contained within the margins and will not amount to range errors. When a baseline falls outside the gating window, we usually wait to improve the baseline naturally. No cranio-caudal corrections were made to the couch during treatment in this study.
- Proton therapy can be a viable option to large HCC because of possible dose escalation without excessive liver toxicity as demonstrated in this study and others, providing the opportunity of conversion therapy and bridge therapy after significant downstaging. It would be interesting to know if there were, and how many HCCs who were initially considered medically inoperable, were eventually converted to resectable. This information may provide important rationale to substantiate the administration of proton therapy to HCC.
Answer: We agree with your comment. We believe that proton therapy will be a new multidisciplinary treatment for large-sized liver cancer. In this study, there were two cases with portal vein thrombosis that led to liver transplantation after proton therapy. It is believed that proton beam therapy for HCC can suppress the progression of the disease, maintain a good general condition, and lead to transplantation.
Minor comments:
- P2, line 59-60. “Furthermore, breathing changes the position of the tumor caused by movement of the liver, …” grammar problem
Answer: We have changed this sentence as follows:
“Furthermore, breathing changes the tumor position in the liver, …”
- P2, line 64, can delete “Therefore”
Answer: We have changed this sentence,
- It said in p2. line 88-89 “The radiation treatments were designed using a radiation treatment planning system with a proton pencil beam algorithm” it seems to suggest that the proton therapy was delivered by pencil beam scanning technique. It would be important to specifically describe the type of proton therapy delivery technique was used.
Answer: We have changed this sentence as follows:
The treatment plan was made using a proton treatment planning system(XiO®-N, Elekta Corp.Stockholm, Sweden) in which the proton dose calculation was performed based on the pencil beam algorithm, and the proton dose distribution was formed to the target shape based on the passive scattering method using the patient's collimator or multi-leaf collimators and patient's bolus.
- In Table 2, it is recommended to the move Eso/stomach/duodenum/colon on the left column to the right column, e.g., Eso (5); stomach (10); duodenum (7), and so on.
Answer: We have revised the table.
- Abbreviation “LC” liver cirrhosis in Table 1 was also used for local control. Consider abbreviating local control to local tumor control (LTC) to avoid confusion to the reader.
Answer: We have changed LC to LTC.
Round 2
Reviewer 1 Report
#4. In Table 4, please provide the number of patients for each categorization.
Answer: We added the number of patients for each categorization to Table 4.
--> New comment: It seems Table 4 was not revised. Please check.
Author Response
We thank referees for careful reading my manuscript and for your helpful suggestions.

Reviewer 2 Report
I was asked to review the revised manuscript about the long term results of PT of HCC. The manuscript is clearly improved. However, I still discerned some shortcomings of the description:
· Abstract: “less invasive” is mentioned twice. “less” implies a comparison which is not explained in the abstract. I suggest to use rather terms like “marker-less” and “breath-hold”.
· P. 3, line 104: the description of the ITV is still disconnected from the one of the breath-hold technique. It seems that the active breathing cycles did not contribute to the ITV margin.
· P. 5, line 183: I don’t get the meaning of the new paragraph. While the first part seems to describe clinical decisions made in the past, the second part seems to describe a dose de-escalation scheme. The word “Therefore” appears to be not appropriate.
· Discussion, page 2 (numbering?), new part, e.g. line 352. Do all numbers refer to liver segments? If yes, why did you use “VI” and 6 for the same segment? In my first revision I asked to elucidate the x-ray alignment with respect to vertebrae and the diaphragm, which refers to the scenario of an anatomical distortion with respect to the planning CT. The new part only describes the uncertainty of the diaphragm-based alignment and disregards the geometrical relation between vertebrae and diaphragm. The reply to the reviewers mentions, e.g., inter-fractional baseline shifts and “the internal margin was customized based on the amount of respiratory-induced motion visualized in the gate width for 1 s”. This is much clearer (although not as clear as desirable) than the description in the manuscript.
Author Response

(The authors gave the same response as above.)

Reviewer 3 Report
The reviewer is thankful for the authors' effort to improve the results according to the feedbacks which are satisfactory.
There are still minor comments that haven't been addressed:
1. abbreviation of local control was not changed to LTC, to avoid confusion with LC (liver cirrohosis).
2. Table 2, Eso/stomach/etc. to be moved to the right column for better readibility.
Otherwise, this manuscript provides important clinical outcomes from proton therapy to large HCC and insights into potential direction of EBRT, specially proton therapy in the multi-disciplinary management of HCC.
Author Response

(The authors gave the same response as above.)
